# Dental Teacher Feedback and Student Learning: A Qualitative Study

**DOI:** 10.3390/dj11070164

**Published:** 2023-06-30

**Authors:** Peter Fine, Albert Leung, Ingrid Tonni, Chris Louca

**Affiliations:** 1UCL Eastman Dental Institute, London WC1E 6ED, UK; albert.leung@ucl.ac.uk; 2Department of Medical and Surgical Specialties, Radiological Sciences and Public Health, Dental School, University of Brescia, 25121 Brescia, Italy; ingrid.tonni@unibs.it; 3Dental Academy, University of Portsmouth, Portsmouth PO1 2QG, UK; chris.louca@port.ac.uk

**Keywords:** feedback, qualitative, dental education

## Abstract

Introduction: Feedback is essential to improve student learning and motivation and to encourage curriculum development by teachers. This study looked at feedback to and from dental students from a qualitative perspective. Methods: Dental teachers were recruited exclusively to this study from the membership of the Association for Dental Education in Europe (ADEE). Delegates from each of the four annual ADEE conferences were invited to attend focus groups to discuss aspects of feedback. Focus groups established an individual theme for the respective conferences: (i) the role of the teacher in delivering feedback; (ii) feedback from the students’ perspective; (iii) changes to feedback due to the COVID-19 pandemic; and (iv) integrating feedback with assessments. Results: Qualitative data collected from the conference delegates were diverse and thought provoking. Delegates reported different styles of feedback varying from individual, personal feedback to no feedback at all. An enforced and mostly positive adaptation to online delivery during the COVID-19 pandemic was reported. A partial return to pre-pandemic practices was described. Conclusions: Feedback is well recognized by students and teachers as contributing to learning. A universal approach to delivering feedback as part of the student learning process can be challenging due to a multitude of variables. Many aspects of changes in dental education, teaching, and feedback practices adopted as a result of the COVID-19 pandemic have been retained post-pandemic, thereby accelerating the anticipated progression to online teaching.

## 1. Introduction

The role of feedback has been well established in general educational terms [1,2], as well as in clinical medical [3] and dental [4,5] undergraduate and postgraduate education. The use of feedback in education is complex, needs to be continuous, and often generates new learning [1,6,7]. The receipt of feedback in the continuous learning of medical/dental students is an integral part of the development of both undergraduate [8,9] and postgraduate students [4,10] and can be considered an essential part of lifelong learning (Murdoch-Eaton and Whittle 2012) [11]. This is fundamental to the individual developing motivation to explore lifelong learning opportunities, thus supporting the principle of “whole person learning” [12], which is the desired outcome for medical and dental students.

In clinical environments, delivering feedback to students can be particularly challenging and multi-faceted, as reported by Fredette et al. (2021) [13].

The primary role of feedback delivered by teachers to students is to enhance student learning [14,15]. Similarly, feedback from students to teachers is also essential to improving teaching and allowing the positive evolution of both the curriculum and the department [16,17]. Feedback can also have a profound impact on student and teacher motivation. Orsini et al. (2018) [18] reported that in a self-sufficient learning environment, the quantity and quality of feedback were positive predictors of students’ autonomous motivation. Supporting student autonomy, competence, and future learning may optimize motivation and facilitate clinical education, leading to a strong predictor of learners’ educational achievement [19]. Communication between student and teacher is fundamental to feedback, involving a degree of social interaction [20].

A two-way approach to feedback [18] contributes to a close relationship and facilitates a good learning experience for both student and teacher. In clinical environments, dental students have reported that, “tutors can make or break your time at Dental School” [21]. Similarly, medical students considered that “multiple meaningful learning activities and individual supervision with continuous feedback” was an essential part of a “good research environment” [22].

Student expectations with respect to feedback vary enormously and are dependent on several variables including cultural influences [23], the use of technology [24], the mode of feedback delivery [4], and changes in the medical/dental curriculum [25,26].

Students are often confused by what feedback is and when it is being delivered. Definitions of feedback are varied and not specifically aimed at the bespoke teaching/learning of medical/dental students. It is suggested that “Feedback is an essential part of education and training programs. It helps learners to maximize their potential at different stages of training, raise their awareness of strengths and areas for improvement, and identify actions to be taken to improve performance” [27]. In educational settings, feedback has been characterized as “Information provided by an agent regarding aspects of one’s performance or understanding” [1]. Feedback has also been defined as “The means by which a student is able to gauge at each stage of the course how he or she is going in terms of the knowledge, understanding and skills that will determine his or her result in the course” [28]. A more recent definition of feedback espoused by Leung et al. (2022) [29], specifically in the dental context, stated “The provision of specific information comparing the clinical and non-clinical performance of students and tutors, against recognized and agreed good practice standards, with the intention of improving this overall performance”.

There has been a huge change in the delivery of dental education throughout the world as a result of the COVID-19 pandemic [30], resulting in significant changes to feedback between teachers and dental students. However, the importance of effective feedback has remained constant, and it is the cornerstone of competency-based education [13].

## 2. Objective

This study aimed to investigate feedback culture and practices in dental education using a qualitative research approach. We chose to look primarily at dental teachers’ perspectives of the feedback they deliver to and receive from students, as well as themes emerging among dental teachers in Europe.

## 3. Methodology

This study sought to collect qualitative data from dental teachers throughout Europe. Qualitative data were collected using (a) questionnaires (delivered online using Google Forms^®^) employing open questions; and (b) semi-structured focus group discussions at four consecutive annual conferences of dental educators in Europe (2018–2022). The first two and the final conference were held in-person, and the third conference was virtual due to the COVID-19 pandemic. The first meeting considered the role of the teacher in delivering feedback, the second looked at feedback from the students’ perspective, the third meeting considered changes to feedback due to the pandemic, and the final meeting looked at integrating feedback with assessment from the teachers’ viewpoint. The questionnaire, which was not validated, was designed by the researchers and was influenced by previous medical questionnaires following the work of Artino et al. (2014) [31]. The questionnaire was distributed by the ADEE secretariate using Google Forms. The targeted teachers were the faculty members of all ADEE member schools and as such, no specific sample size calculation was practicable. The questions posed during the four focus group meetings are detailed in Table 1.

Each focus group discussion was facilitated by a research team member, who was experienced in facilitating focus groups and collecting qualitative data. Discussions between the facilitators were held before the focus groups to calibrate the questions. Conference delegates attending each Feedback Special Interest Group (SIG) were randomly allocated into four focus groups. Each focus group discussed individual themes for 1 h before delivering a report to the overall SIG. Qualitative data were recorded by the focus group facilitator in the form of handwritten notes and direct quotes.

The themes within the questionnaires completed by the conference delegates are detailed in Table 2.

Thematic and narrative approaches to the analysis of the data were employed (Kiger and Varpio, 2020; Polkinghorne, 1995) [32,33].

Ethics Committee approval was granted by University College London, Research Ethics Committee (Project Ethics Identification Number 6552/001).

## 4. Results

Qualitative data were collected from both questionnaires and focus groups generated from four conference meetings held between 2018 and 2022. The merged data were thematically analyzed using a selection of themes and sub-themes generated from the data (see Table 1). Delegates (*n* = 137) from 43 European countries and some nations outside of Europe attended the focus groups. When scrutinizing the datasets for all four meetings, the authors decided to use different approaches to qualitative data analysis to best present the respective data. Data from meetings 1 and 2 were analyzed using a simple inductive thematic approach to best present these data. Data from meetings 3 and 4 reflected teachers’ opinions, concerns, and beliefs; therefore, a narrative approach was adopted.

The number of attendees at the special interest group, the number completing the questionnaire, and the number attending the focus group discussions during year 3 were lower than in other years due to the COVID-19 pandemic; the meeting was facilitated online. This meeting held in 2022 involved three focus group discussions. The small amount of qualitative data in the second questionnaire was not deemed suitable for analysis. No students were invited to attend the focus groups; therefore, only quantitative data were available from them, as previously reported [30]. Qualitative data were taken from three questionnaires and four focus group meetings between 2018–2022.

Diverse themes regarding feedback were identified from the qualitative data collected from comments in the questionnaires and the focus group discussions (Table 3).

Style of Feedback:

Different styles of feedback given by the teachers were identified, including constructive, individual or group, and reflective. The style of feedback delivered by some teachers was consistent and standardized irrespective of the student teaching scenario.


*“Students do not reflect because they do not know what reflection is—reflection is not just ‘thinking in the bath”*



*“When giving clinical/skills feedback, we do not praise the student: they cannot afford to be complacent: we need to point out to them where the problems are. They need to know, and they need to learn in no uncertain terms”*



*“In summative assessments (e.g., OSCEs), we give group feedback only because it is fairer and less complicated; because this is less controversial from the students’ point of view; by not criticising an individuals for their ‘poor’ performance, we run into less problems with appeals against the results/marks awarded”*



*“Our students warm to written feedback, because it feels more comforting, particularly if they have passed”*


Theme:—Student type:


*“There is a difference in undergraduate and postgraduate feedback”*



*“********** [name of a country] students only want grades! They don’t necessarily want feedback”*



*“We only give oral feedback to failing students”*



*“Student character is an important factor in how they receive feedback”*



*“The student needs to act on feedback for it to be effective”*


Theme—Receiving and Delivering Feedback:


*“Our students like written email feedback and so do our teachers, because, like it or not, we need to give some feedback, and it is HR [Human Resource] driven, and so the less interaction there is with students when feedback is given, the better it is for all parties”*



*“Sometimes as teachers we are at a loss as to how to deliver feedback”*



*“Standardizing how we give feedback is ‘mission impossible’”*



*“We can assess whether the student heeds more constructive or encouraging feedback during the discussion to adapt feedback to individual*
*”*


Theme—Professionalism:


*“When delivering feedback, you need to be careful of the language you use and not in front of patients”*



*“In 20 years’, time what will happen to feedback practices, there will be 360 degree feedback”*



*“We as teachers are charged with enlightening our students: the more they know about the limitations of their knowledge and practice the better”*



*“As a tutor you need to understand what the student hasn’t done properly in the first place”*



*“Consistency. Standardization”*



*“Need to adapt your message to the students’ mind-set”*


Theme—Technology


*“Digital technology will allow students to self-evaluate: they will know where the mistakes are without the teachers telling them”*


Meeting 2:

A different thematic analysis was undertaken for meeting 2 (aee Table 4).

Developing Feedback Skills:


*“Following summative assessments, the teacher should go through the exam questions with the students and give good voluntary feedback”.*



*“Face to face feedback needs to be really structured, otherwise you divert and run away from the real issues”.*


Use of technology:


*“I have introduced a new element in my histology course where we use virtual microscopy. Students deliver group responses to which feedback is given online”*


Reflection:


*“Students are getting involved more in reflective feedback”*



*“Reflection better from postgraduates”*



*“Assessment of reflection is difficult/inappropriate”*



*“Students do not reflect because they do not know what reflection is—reflection is not just “thinking in the bath”*


Delivery mode:


*“I deliver feedback orally after receiving oral or e-mail feedback by students”*



*“The type of feedback given is tailored (as much as possible) to the personal needs of the students (notable from an “emotional” stand point)”*


Responsibilities:


*“I ensure I tell students “Let’s do feedback” after clinical session so they recognise they are getting feedback”*



*“Feedback has changed the approach to teaching it is more now student-centred learning”*


Expectations:


*“We are working with a psychologist to guide student instructors”.*



*“setting out expectations for students leads to feed forward”*


Meeting 3:

This meeting was held entirely online due to the European COVID-19 pandemic lockdown. Six delegates attended each focus group.

Focus Group 1 considered the style of feedback and reported that, when possible, clinical teaching was the same as before the pandemic, i.e., their schools had carried on “as normal”. It was emphasized that small group teaching was important, and this led to motivating students to develop self-directed learning skills. This was certainly the case when problem-based learning was discussed. It was reported to be important to continue student dialogue to prevent problems; thus, feedback is needed to help students implement their new knowledge.

Focus Group 2 investigated the different ways of delivering feedback. Teacher participants reported that for formative assessment, the feedback needed to be student-centered and student-led, as this generated a more positive outcome. Greater engagement with students was reported when discussions took place with teachers concerning their actual numerical scores after an assessment. Some tutors reported that the computer screen had appeared to be a barrier for some students, particularly with respect to immediate feedback. Summative assessments—for example, marking and supplying feedback from an essay—were thought to be time-consuming. There was a perceived need for students to reflect more after feedback delivery, particularly feedback following teaching via simulation. The opportunity for students to discuss their results was considered important and a significant step towards reflection. Delegates questioned whether marking rubrics had been changed/modified following changes to assessment during the pandemic.

Focus Group 3 investigated the impact of changes in feedback given to students during the pandemic. Participants reported that many dental schools had made allowances for the pandemic in their approach to feedback. Some students were reported to have suffered economic challenges, causing increased levels of stress. Despite these challenges, it was reported that a significant number of students agreed that blended learning approaches were more effective.

When considering the amount of feedback delivered, some participants reported that feedback had been enhanced/intensified, whilst others reported either no change or less feedback being delivered during the pandemic. There was a general feeling that feedback on an individual level had intensified. The use of technology increased, with participants reporting greater use of social media (WhatsApp) and email to compensate for reduced face-to-face delivery. Students indicated that they felt well-prepared for their careers ahead, were satisfied, and that their tutors had “done a great job” considering the challenging situation.

Focus Group 4 reflected further on the role technology played in the delivery of feedback to students. Communication platforms such as Zoom proved to be useful for delivering feedback to students, particularly for smaller groups. The use of full PPE for students in patient clinics created unique challenges for teachers related to difficulties in recognizing individual students. A commonly held view of the participants was that despite access to new technologies, the delivery of feedback outside the clinic and practical classes was onerous. One participant stated that “*You can’t replace hands-on teaching*”, and that “*Technology can’t replace teaching clinical skills*”.

## 5. Meeting 4

Group 1 considered the importance of good communication at the height of the COVID-19 pandemic and the need for rapid changes in feedback and assessment. Participants felt it was important to take a more reflective approach before making permanent changes to feedback and assessment practices, and to *“stay calm and check the next year’s responses”* prior to making changes. Although some participants reported that their students provided adverse feedback about newly installed assessments as a result of the pandemic, many teachers used this as an opportunity to explore innovative types of assessment. Participants reflected on the need for feedback to be positive in order to achieve a secure relationship between teachers and students.

Group 2 identified contrasting approaches to feedback from different countries relating to changes made to assessment methods, to what degree these were retained, rejected, or modified, and how teachers portrayed different online personae. Participants expressed concerns about the move towards open-book assessments of students’ clinical knowledge. One such concern regarded the increased risk of cheating in such assessments, and the consequent need for closer student monitoring proved to be challenging. The move to online learning stretched student expectations and increased teacher workload. In addition, some participants questioned the level of student engagement and attendance when using online platforms, resulting in increased stress for teachers. It was agreed that students needed to take responsibility for their learning whilst using online platforms. Most participants reported a return to previous face-to-face teaching; however, some new assessments developed for online purposes would be retained.

As for Group 3, as a result of the need for rapid changes to student assessment and feedback, tensions were often identified between university senior management and teachers. Staff development and training were considered positive ways to compensate for teachers’ lack of knowledge and experience of these new technologies employed. The importance of standardizing feedback and assessment was recognised by the participants. Suggestions were made to improve the feedback processes, including the following: separation of evaluation and feedback; external review of the feedback processes; peer review of teaching and feedback; and calibration of teachers giving feedback and evaluating student feedback.

## 6. Discussion

Much research in medical and dental education has used quantitative methodologies [34]; however, there is a trend towards the use of qualitative methods [35], which provide triangulated and in-depth perspectives of ideas, perceptions, and concepts, reflecting the nuances of pedagogies in clinical practice. This allows pertinent issues to be identified, honed into, and explored, in addition to the quantitative analysis of data [36,37,38]. This study is one of the few to emphasize the qualitative aspects of feedback between dental students and their teachers throughout Europe and beyond.

A variety of participants (*n* = 137) from 40 countries contributed to this study, including 28 European nations. Analysis of the results has highlighted the varied use of feedback in dental education. The diverse opinions expressed indicate that teachers fully understand the importance of feedback and its role in student learning. This agrees with Hattie and Timperley (2007) [1]; Higgins, Hartley, and Skelton (2001) [39]; and Keung Hui et al. (2021) [40]. The link between feedback and ongoing learning has been well established [1] for feedback following both face-to-face (Higgins, Hartley, and Skelton, 2001) [39] and virtual teaching [40]. However, a minority of teachers in this study did not agree that feedback contributes to student learning, as corroborated in an earlier study [41].

This study identified student reflection and its role in ongoing learning, following feedback from teachers, in agreement with Devi et al. (2012) [42]. This study also highlighted the value of teaching students how to reflect, concurring with Clynes and Raftery, (2008) [43]. The value of feedback through reflection was also considered to be of importance, particularly for postgraduate students, supporting the observations of Fine et al. (2019) [44].

The COVID-19 pandemic forced medical [45] and dental [46] education providers to change their methods of assessment and feedback by adopting online approaches. This resulted in perceptions of uncertainty amongst some teachers and students [47]. Participants in the current study had mixed views about the efficacy of these changes, with most wanting a return to pre-pandemic formats, citing dentistry as a practical, clinical subject, requiring face-to-face contact (often with patients) [48]. However, participants also noted the importance for dental schools to selectively continue to use the most effective technologies in dental education and thereby to enhance their preparedness for pandemics in the future [49].

The use of marking rubrics during the pandemic was considered to be of greater importance than before because it facilitates a standardized and consistent approach to marking, particularly following the enforced changes to assessment approaches. Cockett and Jackson (2018) [50] also considered that student self-assessment, self-regulation, and understanding of assessment criteria were enhanced by using rubrics. However, students have also reported that rubrics could be restrictive and stressful [51]. Others have also observed a strong correlation between medical student stress and the transition to online learning and assessment [51].

Focus group participants in this study considered the need for students to rapidly become self-directed learners, in agreement with Mukhtar et al. (2020) [52]. Self-directed learning also facilitates student lifelong learning and confidence [53,54]. Other authors reported the importance of co-creation between medical and dental students and their teachers [55], further contributing to more robust self-directed learning. The needs for small group teaching [56] and for students to develop reflective skills [57] were reported. The question of whether more teaching of dental students should take place online is debatable and ongoing (Khalaf et al., 2020) [58].

Although medical and dental students are more familiar with traditional face-to-face learning, the pandemic accelerated the progression towards e-learning; teachers were challenged to swiftly develop e-learning skills and to support their students to use multimedia efficiently and effectively [59].

This study demonstrated a mixed picture, with some dental schools closing their doors to patients and students during the pandemic and others staying open. However, what is indisputable is that dentistry cannot be taught fully online, and that face-to-face feedback given to students during and after clinical sessions is paramount for their ongoing learning and development. These disparities during the pandemic led to greater stress amongst both teachers and students, in agreement with others [60]. Medical and dental students have been shown to have suffered from stress during the pandemic with specific challenges including an uncertain future, academic inadequacies and delays, and personal and family health concerns. A lack of motivation, increased frustration, and loneliness were also reported by students [61]. This study revealed teachers’ concerns with how students coped with online learning, particularly concerning poor skills, lack of interest, insufficient time management, and poor communication amongst medical and dental students [62].

Whether teaching and feedback practices should revert to pre-pandemic approaches is debatable. Some of the new practices and technologies employed will undoubtedly help to improve (i) feedback to students; (ii) reflections amongst students; (iii) online methods for feedback to students; and (iv) student engagement. From the teachers’ perspective, there has been a need to develop new skills in delivering feedback, skills which need to be honed and, in some cases, learned from scratch [63]. There is also the need for a safe environment [64] in which to deliver and receive feedback [65], whether it might be delivered online or face-to-face.

## 7. Conclusions

Feedback is well recognized by students and teachers alike as an integral part of learning. Many of the changes in dental education and teaching practices enforced by the recent COVID-19 pandemic may be permanently adopted. Reverting to pre-pandemic practices was viewed by most teachers as unlikely. Due to a multitude of variables, it has proved to be impossible to develop a universal approach to delivering feedback. However, it is clear that students depend on feedback from teachers to facilitate their learning, and teachers also depend on feedback from students to improve their teaching.

## Figures and Tables

**Table 1 dentistry-11-00164-t001:** Themes discussed during focus groups.

Conference	Focus Group Themes
Conference 1	(i) Type of feedback currently given to students by their clinical teachers/lecturers in different European Dental Schools; (ii) Staff and student perceptions of the delivery, type, and quality of feedback (identifying differences in feedback practices, perceptions, and rationales); (iii) Common core values and teaching and learning principles to develop more effective feedback practices; (iv) A consensus on feedback approaches in Europe identifying good practice.
Conference 2	(i) Feedback following summative assessment; (ii) Feedback following clinical activity; (iii) Feedback following laboratory-based teaching; (iv) Feedback using technology.
Conference 3	(i) Changes instigated following the COVID-19 pandemic; (ii) Evaluation of changes following the COVID-19 pandemic; (iii) Determination of whether the COVID-19 changes are permanent.
Conference 4	(i) Changes in student assessment strategies following feedback from students/teachers/patients; (ii) Changes in student assessment during the COVID-19 pandemic and whether they are accepted as permanent; (iii) Measures adopted to ensure teachers are kept up to date with assessment approaches.

**Table 2 dentistry-11-00164-t002:** Summary of attendees at each conference, number of questionnaires completed, and questions asked on all four questionnaires.

Conference	Number of Attendees/Focus Group	Questionnaires Completes/Meeting	Focus Group Themes
Conference 1	43 Teachers	172 Teachers	(i) Type of feedback currently given to students by their clinical teachers/lecturers in different European Dental Schools; (ii) Staff and student perceptions of the delivery, type, and quality of feedback (identifying differences in feedback practices, perceptions, and rationales); (iii) Common core values and teaching and learning principles to develop more effective feedback practices; (iv) A consensus on feedback approaches in Europe identifying good practice.
Conference 2	39 Teachers	234 Students	(i) Feedback following summative assessment; (ii) Feedback following clinical activity; (iii) Feedback following laboratory-based teaching; (iv) Feedback using technology.
Conference 3	15 Teachers	67 Teachers	(i) Changes instigated following the COVID-19 pandemic; (ii) Evaluation of changes following the COVID-19 pandemic; (iii) Determination of whether the COVID-19 changes are permanent.
Conference 4	40 Teachers	240 Teachers	(i) Changes in student assessment strategies following feedback from students/teachers/patients; (ii) Changes in student assessment during the COVID-19 pandemic and whether they are accepted as permanent; (iii) Measures adopted to ensure teachers are kept up to date with assessment approaches.

**Table 3 dentistry-11-00164-t003:** Thematic analysis for meeting 1.

Theme	Sub-Theme	Sub-Sub-Theme
Style of Feedback	Reflective Constructive Negative Groups Individuals Standardized feedback Peer review Verbal, written, and video	Staff Students
Type of student	Different year groups Feedback only for failed students	
Receiving/delivering feedback	Increased learning Correct interpretation Improved quality of assessment Individual adaptation of feedback Feedback on feedback Patient feedback Feedback with/without grades	Not in front of patient Avoid low grades
Technology	Digital Future developments	Self-assessment Clinical/Academic evaluation Patient records Video/Audio clips Digital learning platforms Recording clinics/Body cameras 360° recordings
Professionalism	Variations between academic years Students to act on feedback	

**Table 4 dentistry-11-00164-t004:** Thematic analysis of meeting 2.

Theme	Sub-Theme
Developing feedback skills	Tutor learning Student learning Consistency of tutors Group vs. individual feedback
Use of technology	Different platforms Use of videos
Reflection	Self-assessment Undergraduates/Postgraduates
Delivery mode	Peer review Individual
Responsibility	Tutors Students
Expectations	Tutors Students Use marking guides

## Data Availability

Data availability is through the corresponding author P.F.

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
