# Peer review of "Dental Teacher Feedback and Student Learning: A Qualitative Study"

_dentistry, 2023, doi:10.3390/dj11070164_

Round 1
Reviewer 1 Report
An interesting paper, with data that may add somewhat to our knowledge in this area. However, the scope is currently too broad and the manuscript would greatly benefit from reduced length and greater focus. There has been a valiant attempt to use the pandemic as an opportunity to expand the scope of the research but in my view this detracts from, rather than adds to, the study. The approach to analysis of the data derived from meetings 1&2 and 3&4 differs in a way that has not been adequately justified. Furthermore, as the authors show, the number of participants for questionnaire and meeting 3 is smaller that that for the other meetings. In my view the authors could present and discuss only the data from the first two questionnaires/meetings.
The lack of focus begins with the objective, which is too vague. In the methodology section, Tables 1 & 2 describe the themes explored in both questionnaires and meetings. In a sense, these themes "reveal" rather more specific objectives for the research, which could be more explicitly expressed in a statement or list of aims.
Since Table 1 describes the themes for the focus groups and Table 2 the questions in the questionnaire, I would suggest that they are reversed in order, to reflect the order in which the collections occurred. Some comment is required about how and to whom the questionnaires were distributed, and the timing of this in relation to the focus groups. Were the questionnaires analysed to then inform the themes for the focus groups? If not this seems like a missed opportunity. Further comment is also required in relation to the format of the focus groups. Were they semi-structured? The authors state that the facilitators were experienced, but that does not remove the need for some form of calibration. I would also prefer to see more information on the thematic analysis. Was this inductive, deductive, both?
Results - there would appear to be 2xTable 3. Examples are given of comments under each identified theme but it would be interesting to understand if any views were more frequently expressed than others. The quotations in lines 178-183 are not in italics. The quotation in lines 197-198 is a repeat of one provided earlier. In relation to meeting 2, there are no verbatim examples of data in relation to "developing feedback skills", which seems a shame because this is an important issue.
For me the introduction doesn't create a compelling case for why this research is necessary, which I am sure it is. What questions do we need answers to that only a qualitative approach can provide? That needs to be clearly drawn out of the published literature. The discussion then needs to focus on how the data adds to our understanding. An obvious message from the data is the lack of consistency and practice across the participants and, one might surmise, national groupings. More interpretation of the data, if justified, would be helpful, perhaps with suggestions for how more evidence-based practice might be encouraged. A more in-depth treatment of these issues would be possible, and valuable, if consideration of the effects of the pandemic were removed. The impact of the pandemic on dental education has received attention elsewhere and, unfortunately, there are questions about the robustness of the analysis of the pandemic-related data presented in this study.
Reviewer 2 Report
This is a very interesting study done to assess the impact of feedback on student learning and motivation. The data is well collected and extensive and analysis is presented well.
Some minor feedback below:
-Please ensure Table 2 and Table 3 start on a new page, on my printouts they started at the bottom of the page and carried over to the next page.
-Line 178-183 are quotations and should be in italic.
-Line 148-149 and 197-198 are identical comments. Please confirm if this is intentional or an error.
-Capital S at start of line 209.
Reviewer 3 Report
This study aimed to evaluate the feedback to and from dental students from a qualitative perspective in a group of ADEE members and delegates throughout Europe.
The study is well planned and in general suitable for publication. The reviewer thinks that the topic would be more suitable for a Journal with focus on Dental education.However, some issues need to be addressed prior to considering this paper for publication:
Abstract: It´s not clear who participated in the study: teachers only or also students?
Introduction: this section is very lengthy and should be shortened. In contrast: the objectives (line 97-99) are very short and need more explanation.
The information in table 3 may be included in either table 1 or 2. In general, it´s not very clear in the methodology who the participants were. This should be very clear at the beginning.
Possibly not all European countries participated in the study. This might have led to a selection bias in the answers, also the conclusion may not fit for all countries. A list with number of participants of the participating countries would be useful to the reader.
Reviewer 4 Report
Dear authors,
With interest I’ve read the paper “Dental Teacher Feedback and Student Learning: A Qualitative
Study”.
The study is well-planed and thoroughly presented. It is of interest to a wide range of readers among dental professionals. However, some comments are addressed.
Abstract
The abstract seems to bee too ling. Please, check the instructions for the authors.
At least some information about the questions in the questionnaire should be given.
Results should be presented in the abstract in more details, not just in general words.
No discussion section should be present in the abstract. There should be a short conclusion instead.
Keywords: please, check if the key-words are MeSH terms.
Introduction
Please, check the reference format.
Methodology:
The methodology section should be extended.
Who and how defined the questions in the questionnaire?
How was the questionnaire distributed?
Was the questionnaire validated in some way?
Was the target number of participants assessed?
How the qualitative analysis of the teachers answers was accomplished? How the quantitative analysis of the student’s answers was accomplished?
Results
Results are presented in very unclear way. It seems the whole section should be rewritten.
Discussion
Comparisons with previous quantitative and qualitative studies are essential.
Minor editing of English language required
Round 2
Reviewer 1 Report
None
Editing required to correct typographical errors.
Author Response
Thank you to reviewer one for pointing out the anomalies, which have now been corrected. With respect to background and references in the introduction we have reviewed the introduction and feel that it is comprehensive, up to date and well-referenced.
The methods have been similarly reviewed and reflect current approaches adopted in the literature.
The presentation of results have already been modified. The nature of qualitative data reporting requires a heavily worded approach, which is often not standardized.
We have reviewed the conclusion and feel that they adequately support the results.
Reviewer 3 Report
The authors addressed the reviewer´s comments sufficiantly.
Author Response
Thank you to reviewer 3 for their positive comments and time.
Reviewer 4 Report
Dear authors,
I am satisfied with the changes made.
Minor grammar and spelling editing required.
Author Response
Thank you to reviewer 4 for their feedback. All minor editing and grammar issues have been addressed.
With respect to the results, they have been further reviewed and checked to ensure that they are clearly presented.